# Transcriptional Factor Repertoire of Breast Cancer in 3D Cell Culture Models

**DOI:** 10.3390/cancers14041023

**Published:** 2022-02-17

**Authors:** Hande Özkan, Deniz Gülfem Öztürk, Gozde Korkmaz

**Affiliations:** 1School of Medicine, Koç University, Istanbul 34450, Turkey; hozkan18@ku.edu.tr; 2Research Centre for Translational Medicine (KUTTAM), Koç University, Istanbul 34450, Turkey

**Keywords:** breast cancer, tumor heterogeneity, three-dimensional (3D) culture, transcription factor, extracellular matrix

## Abstract

**Simple Summary:**

Knowledge of the transcriptional regulation of breast cancer tumorigenesis is largely based on studies performed in two-dimensional (2D) monolayer culture models, which lack tissue architecture and therefore fail to represent tumor heterogeneity. However, three-dimensional (3D) cell culture models are better at mimicking in vivo tumor microenvironment, which is critical in regulating cellular behavior. Hence, 3D cell culture models hold great promise for translational breast cancer research.

**Abstract:**

Intratumor heterogeneity of breast cancer is driven by extrinsic factors from the tumor microenvironment (TME) as well as tumor cell–intrinsic parameters including genetic, epigenetic, and transcriptomic traits. The extracellular matrix (ECM), a major structural component of the TME, impacts every stage of tumorigenesis by providing necessary biochemical and biomechanical cues that are major regulators of cell shape/architecture, stiffness, cell proliferation, survival, invasion, and migration. Moreover, ECM and tissue architecture have a profound impact on chromatin structure, thereby altering gene expression. Considering the significant contribution of ECM to cellular behavior, a large body of work underlined that traditional two-dimensional (2D) cultures depriving cell–cell and cell–ECM interactions as well as spatial cellular distribution and organization of solid tumors fail to recapitulate in vivo properties of tumor cells residing in the complex TME. Thus, three-dimensional (3D) culture models are increasingly employed in cancer research, as these culture systems better mimic the physiological microenvironment and shape the cellular responses according to the microenvironmental cues that will regulate critical cell functions such as cell shape/architecture, survival, proliferation, differentiation, and drug response as well as gene expression. Therefore, 3D cell culture models that better resemble the patient transcriptome are critical in defining physiologically relevant transcriptional changes. This review will present the transcriptional factor (TF) repertoire of breast cancer in 3D culture models in the context of mammary tissue architecture, epithelial-to-mesenchymal transition and metastasis, cell death mechanisms, cancer therapy resistance and differential drug response, and stemness and will discuss the impact of culture dimensionality on breast cancer research.

## 1. Introduction

Breast cancer (BC) is characterized by phenotypic and molecular heterogeneity that differs within each patient (intertumoral heterogeneity) as well as distinct genomic and biological features that differ within a tumor (intratumor heterogeneity) [1]. Histopathological classification of BC, referring to its intertumoral heterogeneity, is based on morphological diversity of tumors and contains three main subtypes: ductal carcinoma in situ (DCIS), invasive ductal carcinoma (IDC), and invasive lobular carcinoma (ILC) [2,3]. With the advances of high-throughput gene expression microarrays and next-generation sequencing, multiple molecular subtypes of BC differing by their transcriptional signatures have been identified: Luminal A, Luminal B, Normal-like, HER2-enriched, Claudin Low, and Basal-like/Triple Negative (TNBC) [4,5,6,7,8]. Despite the improvements in BC classification, dissimilar clinical outcomes and changes in survival rates are still inevitable due to the morphological and molecular heterogeneity.

Intratumor heterogeneity of BC typically exists at every level as a consequence of differences in tumor cell–intrinsic factors including (epi)genome [9,10,11], transcriptome [5,6], and proteome profiles [12], intrinsic cell plasticity [13], epithelial-to-mesenchymal transition (EMT), and stemness [8,14]. Tumor heterogeneity is also influenced by extrinsic factors in the tumor microenvironment (TME) originating from the spatial differences of the tumor mass. These factors include tumor hypoxia, the extent of tumor vascularization, host immune response, and interactions between tumor cells and their microenvironment [15]. In more detail, the extracellular matrix (ECM), as a core component of TME, regulates tumor progression by providing necessary biochemical and biomechanical cues affecting cell shape/architecture, ECM stiffness, cell proliferation, invasion, and migration (Figure 1) [16,17]. This is achieved by increased deposition and crosslinking of ECM proteins including collagens, laminins, fibronectin, glycoproteins, and proteoglycans. Hence, ECM remodeling changes the three-dimensional spatial topology and biochemical/biophysical properties of the matrix, thereby affecting the cell fate [18].

A network between tumor cells and their microenvironment, specifically the ECM, was first conceptualized as “dynamic reciprocity”, which refers to continuous bidirectional interactions between cells and the ECM that induce dynamic changes during tissue morphogenesis, organ homeostasis, and malignant cancer progression [19]. These dynamic cell–cell and cell–matrix interactions are achieved by mechanical forces that are exerted by ECM remodeling and by modifying biochemical signals mediated by hormones and cytokines, thereby leading to cellular changes to adapt to different environmental cues. Critically, mechanical signals transmitted from transmembrane receptors to the nucleus regulate chromatin structure and gene expression. Therefore, the ECM directly influences gene expression and in turn forms a feedback loop to modulate its own deposition and remodeling [20,21]. Numerous research studies clearly demonstrate the importance of the ECM and its power in dictating gene expression with three-dimension (3D) culture of mammary epithelial cells and cell lines [22,23,24,25,26].

It is not surprising that cells cultured in 3D have distinct gene expression levels when compared to their two-dimensional (2D) counterparts. Although the traditional in vitro cancer research was based on 2D cell culture using plastic substrates for decades, a large body of work underlined that many cellular features including tissue architecture, proliferation, signal transduction, and cell–ECM interactions are impaired in these non-physiological conditions [27,28] and thus unfortunately fail to completely represent tumor cells residing in the complex TME. The 3D cell culture models are better at reflecting the 3D tissue architecture and shaping the cellular responses according to the presence of microenvironmental cues that will ultimately regulate the cellular behavior and gene expression [29]. In addition to that, these models have the power to decipher the activity of anti-cancer treatments in vitro [30] and bring a new dimension to drug discovery [31]. Therefore, 3D cell culture methodology is more representative of tumor characteristics than traditional 2D monolayers [32]. Table 1 summarizes some of the key differences in 2D and 3D cell culture systems.

Pioneering research by the Bissell group and others showed that human mammary epithelial cells, both normal and malignant, display dramatic morphological and biochemical differences depending on their culture dimensions based on traditional 2D cell culture using plastic substrates vs. 3D cell culture using a laminin-rich extracellular matrix (lrECM) [44]. Although normal mammary epithelial cells and tumor cells are dissimilar in terms of organization and proliferation state in vivo, characteristic phenotypic traits are masked in traditional 2D cultures (monolayers). In fact, when cultured in 2D, normal mammary epithelial cells lose their polarity and exhibit similar plasticity displayed by malignant cells in vivo [45]. However, normal mammary epithelial cells grown in 3D become growth-arrested and inherit the ability to self-organize in a single layer of polarized cells forming small sacs that contain a hollow lumen that corresponds to acinar structures found in vivo. On the other hand, malignant cells cultured in 3D form proliferating, disorganized aggregates without a hollow lumen, lacking both polarity and an organized basement membrane [45]. More interestingly, Weaver et al. showed that the tumorigenic phenotype of human BC cells cultured in 3D, but not in 2D, can be “reverted” into a phenotype of normal-like breast acini with a profound reduction in aggressiveness by antibody-mediated inhibition of β1-integrin, the laminin 1 receptor [46]. Integrins are the main cell adhesion receptors for ECM molecules and mediate cell–cell and cell–ECM interactions [47]. Accumulating evidence showed that integrin-dependent modulation of ECM stiffness and remodelling play vital roles in cancer progression including invasion and metastasis, mediating a cancer stem cell (CSC)-like property and drug resistance [48,49,50]. In addition, altered expressions of integrins are found in a variety of carcinomas, including BC [48]. In addition to β1-integrin-mediated reversion, attenuation of EGFR, MAPKK or PI3K signalling is sufficient to revert breast tumor cells from a disorganized phenotype to growth-arrested and well-differentiated polar acinar structures [51,52,53].

In 3D culture of mammary epithelial cells, ECM-induced cytoskeletal changes lead to deacetylation in histones H3 and H4, which is associated with increased chromatin condensation and reduced global gene expression [21]. Moreover, ECM-induced tissue polarity and acinar formation in mammary epithelial cell line HMT-3522 triggered the spatial organization of several factors including nuclear protein NuMa, the cell cycle regulator Rb, the RNA splicing factor Rm160, NF-κB, and TIN2 [54,55,56]. Furthermore, monitoring the changes by genome-wide gene expression profiling of non-malignant human mammary epithelial cells differentiated into growth-arrested and polarized structures in 3D lrECM culture identified a gene expression signature set that is downregulated during acinar morphogenesis and can be used to predict good prognosis in BC with high accuracy [57]. The signature set includes genes involved in mitosis and the cell cycle (CDKN3, STK6 and EIF4A1), cytoskeletal organization (ACTB), and cell survival (TNFRSF6B).

In their seminal paper of 2007, Kenny et al. presented the first comprehensive analysis of transcriptome profiling, as well as morphological phenotyping, of a substantial panel of BC cell lines in 2D and 3D cultures [58]. The panel consisted of 25 BC cell lines, closely resembling morphologic, genomic, mutational, and transcriptomic diversity of breast tumors in vivo. Affymetrix gene expression analysis of cells that were classified into four distinct morphological classes in 3D culture pointed out that although cell line identity and the Luminal/Basal phenotype greatly influence the gene expression profiles, the culture microenvironment also causes significant effects. Indeed, genes involved in signal transduction and enzyme regulation were significantly overrepresented in the set of genes, which differ between 2D and 3D cell cultures [58]. Another critical study that implies altered expression of gene classes was performed in 3D models of preinvasive to invasive transition in BC. Differentially expressed genes between preinvasive and invasive cells were grouped as integral membrane proteins, transcription factors, kinases, and chemokines. Furthermore, matrix metalloproteinases (MMPs; MMP13, MMP15, and MMP17) had higher expression in invasive cells versus preinvasive cells in 3D cultures. The effect of MMP expression on the invasive phenotype was evaluated with pharmacological inhibition and knockdown of these MMPs, which abrogated invasiveness [59].

Recent evidence highlights the link between differential gene expression profiles and subtype-specific morphological changes by comparing the transcriptome of BC cell lines cultured under both 2D and 3D conditions [60]. Regardless of their subtypes, BC cells cultured in a 3D Matrigel condition showed an increased expression of genes involved in generic transcription and oxidative phosphorylation and a downregulation of cell cycle-related genes, including various cyclin-dependent kinases (CDKs) and aurora kinases. On the other hand, the cell cycle pathway was enriched in 2D cell culture conditions, reflecting the significant increase in growth rate compared to 3D cell cultures. Most critically, they concluded that changes in gene expression patterns in different culture dimensions are subtype-dependent. For instance, expression of the genes involved in cell–matrix interactions, ECM organization, collagen synthesis, and integrin cell surface interactions was induced in 3D for the Basal B subtype coinciding with its invasive phenotype, while decreasing for the Basal A subtype. They further concluded that the transcriptomic correlation between patient tumors and in vitro cell culture models was highly dependent on the BC subtype [60]. Altogether, these pioneering results highlight that employing 3D cell cultures in association with ECM leads to particular and significant gene expression changes.

As reviewed above, it is now well established that interplay between cancer cells and their microenvironment dictates transcription to influence highly complex and divergent intracellular signal transduction pathways, which mostly end up with activation or inactivation of regulatory molecules [61]. Among these regulatory molecules, transcription factors (TFs) act as critical players by combining the extracellular signals with intracellular signal transduction pathways and regulating the expression of target genes that have essential and/or partial functions in key cellular processes including proliferation [62], metastasis [63], inhibition of apoptosis [64,65], and drug resistance [66]. However, what we know about the inference of the TFs in tumorigenesis is largely based on the studies performed in 2D monolayer culture, which fail to imitate TME and ignores spatial cell–ECM interactions.

In this review, we provide an overview of the transcriptional factor repertoire of breast cancer in 3D cell culture models. These TFs are discussed in the context of mammary tissue architecture, epithelial-to-mesenchymal transition (EMT) and metastasis, cell death mechanisms, cancer therapy resistance and differential drug response, and stemness, and their contributions to BC research are underlined.

## 2. Transcriptional Regulation of Mammary Tissue Architecture

The mammary gland, as a highly complex and well-organized tissue, comprises apically orientated luminal epithelial cells and basally orientated myoepithelial cells that come together to form cylindrical ducts. Basal myoepithelial cells are enveloped by a basement membrane (BM) containing collagen IV, laminins, entactin, and proteoglycans. The BM, itself, is embedded in the stroma, which has a cellular composition of many different cell types including adipocytes, endothelial cells, immune cell infiltrates, and fibroblasts [67]. These cells synthesize the ECM components essential for the 3D microstructure of the stroma.

One of the hallmarks of breast tumorigenesis is the disruption of mammary tissue architecture [68]. Under non-tumorigenic conditions, breast epithelial cells constitute polarized and well-organized structures called “acini’’ as the smallest functional units that are responsible for producing milk in the terminal duct [69], and the TME controls the epithelial cell phenotype with distinct cellular signals and regulatory factors. During tumorigenesis, well-established acini structures (healthy cell morphology and polarized tissue structure) are disrupted, and disorganized cells lacking polarity emerge to induce more invasive and aggressive phenotypes (Figure 2) [70,71].

In 3D cell culture models, these complex events are successfully mimicked to specify the cellular signals and regulatory factors that are generated through the TME. Therefore, it is possible to identify and characterize these regulatory mechanisms [72]. In this section, we will summarize the literature on TFs regulating mammary tissue architecture in 3D cell culture models.

The *TP53* gene (tumor protein P53; p53), a nuclear TF, regulates homeostasis through transactivation of numerous target genes that are involved in cell cycle arrest, DNA damage and repair, and apoptosis [73]. It is the most frequently altered and/or mutated tumor suppressor gene in human cancers including BC [74]. Although most p53 mutations cause loss of the tumor-suppressive function, gain-of-function (GOF) mutations of p53 often promote cancer progression, metastasis, and chemoresistance [75,76]. For instance, mutant p53 with GOF is recognized as an important driver of ECM remodeling, which is a key step in metastatic niche priming [77,78]. Intriguingly, exosomes released from mutant p53-expressing cancer cells enhance integrin trafficking in fibroblasts via the Rab-coupling protein (RCP)/diacylglycerol kinase-α (DGKα) pathway, leading to pro-invasive ECM remodeling that could serve as pre-metastatic niches in target organs [78].

P53 is considered a potent candidate responsible for disorganized morphology of BC, which can be reverted when the mutant form of p53 is depleted [79]. Genome-wide expression analysis indicated that mutant p53 coactivates the mevalonate pathway to modify lipid and sterol biosynthesis, which results in atypic phenotypic effects on breast tissue architecture. The mutant p53 interacts with promoters of other TFs (*SREBP-1* and *SREBP-2*; sterol regulatory element binding transcription factor 1 and 2) that activate the genes encoding key enzymes in both fatty acid and sterol biosynthetic pathways [79]. Hence, phenotypic effects of mutant p53 are mediated through its physical interaction with SREBPs, which leads to the upregulation of mevalonate pathway-related genes. Thereby, mutant p53 could directly affect architecture of breast cells through the mevalonate pathway, and reversion of the related mutation in p53 resulted in conversion from disorganized structures into reminiscent acinar structures in 3D [79].

NFKB (NF-κB, Nuclear Factor kappa B), the key TF linking inflammation with cancer, regulates a number of cancer-related processes such as the immune response, cell survival, and cancer progression [80]. Most importantly, it has been identified as a regulator of tissue polarity and organization [81,82], and activation of NFKB in tumors disrupts important microenvironmental signals necessary for tissue organization and leads to the transmission from polarized spheres (‘acini’) into large tumor-like clusters. In 3D culture of BC cells HMT-3522, organized tissue morphology has been altered with NFKB- mediated transcription activation of the disorganization related genes that have potential NFKB binding sites in their promoter region [83], and upregulation of disorganization related genes are positively correlated with an invasive and aggressive phenotype in 3D cultures of multiple BC cell lines. Phenotypic change due to activation of NFKB can be reversed by blocking of EGFR, beta1, integrin, MMPs, or their downstream signals [83].

GATA3 (GATA Binding Protein 3) is a well-known TF with a role in mammary gland development and it is significantly expressed in most estrogen receptor-positive (ER+) tumors [84,85]. It is also stated as one of the pioneer factors that functions upstream of ER alpha (ERα) that forms a positive cross-regulatory feedback loop [86]. Somatic mutations within *GATA3* have been reported in the majority of luminal-type BC, and these mutations are generally associated with uncontrolled proliferation with a generation of novel active GATA3 variants. In 2018, two mutations in GATA3 (GATA3308* and GATA3335fs) affecting the DNA binding domain specifically located in the zinc-finger domain were generated and ectopically expressed in the ZR751 BC cell line [87]. Larger colony sizes in 3D culture were observed even in the absence of estradiol induction (absence of estrogen). This study clearly proves that GATA3 has unique and diverse somatic gain of function mutations, enhancing the proliferation and growth of the BC cell line in 3D culture [87].

Dysregulation of the Hippo signaling pathway, a key regulator of tissue growth and homeostasis, leads to the activation of the TAZ oncogene and its transcriptional target *IRS1* (insulin receptor substrate 1) in BC [88,89]. Taking advantage of the 3D cell culture, the crosstalk between two important tumorigenesis-related pathways, Hippo-signalling with TAZ and insulin signalling pathway with IRS1 had been modeled successfully [90]. Critically, TAZ-mediated IRS1 upregulation induced the proliferative transformation phenotype with a disorganized structure in MCF10A mammary epithelial cells in 3D Matrigel culture.

Moreover, when cultured in 3D, non-tumorigenic mammary epithelial MCF10A cells ectopically expressing Runx2 (Runt-related transcription factor 2) show disrupted acinar structures, enhanced proliferation with a decreased level of apoptosis, and loss of β4 integrin, which is an essential component for the organization of the glandular epithelium [91]. Conversely, knockdown of endogenous *Runx2* or stable expression of functionally deficient mutant versions of Runx2 restored organized acini structures in metastatic MDA-MB-231 in 3D culture and failed to promote tumor growth in vivo [91].

Another TF regulating mammary tissue architecture is *SOX7*, a tumor suppressor gene located on human chromosome 8, which is frequently lost in breast tumors [92]. shRNA-mediated knockdown of *SOX7* in the 3D culture system of MCF10A forms irregular clusters with a lack of polarity [93]. In addition to *SOX7*, MRTF-A and B (Myocardin-related TFs, which are known to link actin-based dynamics and mechanotransduction with intracellular regulatory pathways [94,95], affect proliferation and formation of mammary acini from luminal epithelial cells, as overexpression of these factors resulted in irregular mammary acini structures with increased size, significant defects in lumen formation, and defective apicobasal polarity in MCF10A cells grown in 3D culture [96].

Experimental models of mammary epithelium demonstrate that ECM stiffness, which compromises tissue architecture, can alone induce the malignant phenotypic change in non-malignant mammary epithelium [97,98]. Accordingly, increased ECM stiffness triggers a tumorigenic phenotype with disrupted nuclear morphology and chromatin structure in MCF10A breast epithelial cells cultured in mechanosensitive 3D culture models [97,98]. Interestingly, more accessible peaks were observed for cells cultured in stiff matrices in contrast to soft matrices according to ATAC-Seq, which provides a genome-wide and site-specific view of changes in chromatin architecture [99,100]. These accessible sites frequently present Sp1-binding motifs, and increased activity of the Sp1-HDAC3/8-mediated pathway was found to be critical for the regulation of a stiffness-induced tumorigenic phenotype in their 3D culture model of mammary epithelium. Additionally, Stowers et al. underlined that epigenetic profiles of mammary epithelium are significantly altered by culturing conditions [100]. Culture of MCF10A cells on 3D soft matrices more closely reproduce the in vivo chromatin state of human mammary tissue compared with cells cultured in 2D. Altogether, this work draws attention to the necessity of conducting chromatin profiling experiments and transcriptome analyses in mechanoresponsive 3D cell culture systems [100].

## 3. Transcriptional Regulation of Epithelial to Mesenchymal Transition (EMT) and Metastasis

Solid tumor progression requires the transition from an epithelial to a mesenchymal phenotype, which is characterized by switching off the expression of cell adhesion molecules such as E-cadherin and turning on mesenchymal markers such as N-cadherin [101]. In this transition process, epithelial cells lose both their polarized phenotype and tight cell–cell junctions through different molecular pathways and gain a mesenchymal cell phenotype, which leverages features of invasiveness, increased motility, and resistance to apoptosis to invade and metastasize [102,103]. Several pathways including transforming growth factor-beta (TGF-*β*), tumor necrosis factor-alpha (TNF-*α*), and Notch, Hedgehog, and Wnt pathways can induce EMT by upregulating TFs, which promote re-organization of the ECM [104]. Specifically, ECM remodelling supports the migration and invasion of tumor cells through several different biological events: changes in the composition of the BM, proteolysis-dependent and -independent degradation of ECM, altering cell–ECM interactions, secretion of various pro-fibrotic growth factors and inflammatory factors, and promoting angiogenesis [105]. Overall, remodelling of ECM and subsequent biochemical changes create a cancer-supporting matrix leading to the development of the metastatic character of epithelial cells [106]. So far, the link between the EMT and tumor progression has been mostly studied under 2D culture conditions, which poorly resemble the complexity of in vivo tumors. However, 3D cell culture models incorporate the in vivo elements of tumors, such as complex cell–cell and cell–ECM interactions, which are critical for evaluating the EMT process and metastasis [107,108]. In this section, we will summarize existing literature that mainly implicates TFs regulating EMT and metastasis in 3D cell culture models. Pioneering studies on co-culture systems, which highlight the applicability of 3D culture systems for efficient physiological environment modelling, are also included.

In BC, Twist1 (Twist family bHLH transcription factor 1) overexpression is usually associated with poor prognosis and metastases (lymph-node and distant) [109]. Several studies have demonstrated that Twist1 regulates tumor invasion and metastasis by inducing EMT in 3D cultures of BC [110]. A positive correlation between *Twist1* and *ADAM12* (metalloproteinase 12) mRNA expressions was reported in human breast tumor samples [111,112,113]. Although knockdown of *ADAM12* did not cause any change in the ability of Twist1 to induce EMT in 2D culture, experiments performed in 3D culture and in vivo concluded that ADAM12 is necessary for Twist1-induced invasion and metastasis by promoting invadopodia-mediated ECM degradation [113]. A novel molecular framework for Twist-1 induced epithelial invasion and dissemination in basal BC was identified using primary mammary organoids from a Tet-inducible mouse model, and results were validated both in a metastatic mouse model and in primary patient tumor tissues [114]. In a 3D mammary epithelial culture model, Twist1 directly regulated transcriptional upregulation of *Prkd1* (Protein kinase D1), whose activity promotes ECM-directed invasion, loss of cell–cell adhesion, persistent migration, and metastasis without loss of epithelial identity [115,116]. Consistently, *Prkd1* knockdown reduced invasiveness at the tumor–stroma border and resulted in significantly fewer lung metastases in vivo [116]. Moreover, Twist1 has been identified as a downstream target of SRC-1 (Steroid receptor coactivator-1) that is usually overexpressed in HER2-positive BC. Through its coactivator function promoting PEA3-mediated Twist1 transcription, SRC-1 inhibited E-cadherin expression, formed undifferentiated invasive structures, and promoted BC cell migration and invasion in 3D culture [117].

EMT is a highly conserved biological process to initiate cell invasion and metastasis [118]. The Snail family of TFs is a prominent inducer of EMT and is responsible for conversion from a non-invasive to an invasive phenotype leading to metastasis [119]. Targeted inhibition of Snail reduced the invasive and metastatic propensity of MCF7 cells in 3D culture [120]. Another important factor is STAT3 (Signal transducer and activator of transcription 3), which has been shown to increase migration and invasion of oncogenic v-Src-transformed MCF10A by targeting long non-coding RNA LINC00520 in 3D culture [121]. Upregulation of STAT3 activity through Rac GTPases was also associated with invasion of BC, as proven in 3D primary tumor tissue explant culture [122]. In another study, suppression of TF p63, particularly the i∆Np63α isoform, leads to H-Ras-mediated EMT and cell invasion of mammary epithelial cells in a 3D collagen gel system [123,124,125]. They suggest that HER2 and Ras inhibitor combination may lead to improved clinical efficacy for treatment of HER2-positive BC, which usually develops resistance to Herceptin [126]. A metalloprotease involved in angiogenesis and cancer invasion, MMP14 is negatively regulated by TF PROX1 (Prospero Homeobox 1), which causes a significant reduction in 3D cell invasion of MDA-MB-231 cells. Not surprisingly, *PROX1* is epigenetically silenced in BC to promote MMP14-dependent invasiveness [127].

Metastasis and cancer cell motility are also correlated with phenotypic and genomic changes caused by the NUDT5 (Nucleotide Diphosphate Hydrolase Type 5) enzyme that is commonly overexpressed in BC and is associated with a more aggressive cancer phenotype [128,129]. Analysis of the gene expression changes in BC cells stably depleted of *NUDT5* revealed that expression of stem cell TF complexes (OCT1-SOX2, PAX6-SOX2 and OCT4-SOX2) significantly increased during the transition from 2D to 3D growth conditions [130,131]. Furthermore, SP1, ZNF281, and MZF1 motifs were enriched within promoters of genes involved in the transition between 2D and 3D. Among them, overexpression of SP1 (Specificity protein 1, regulates genes involved in metabolism, cell proliferation/growth, and cell death [132]) and ZNF281 (Zinc finger protein 1, an EMT-inducing transcription factor [133]) is related to poor prognosis in BC. Moreover, MZF1 (Myeloid zinc finger 1), which is known as the oncogenic TF in various solid cancers, has a critical role in trans-differentiation of mesenchymal stem cells (MSC) into carcinoma-associated fibroblast (CAF)-like cells to promote tumor growth and metastases in BC [130,131]. Another EMT-related TF studied in 3D culture models is *ZEB1*, a zinc-finger E-box-binding TF that regulates expression of E- and N-cadherins. *ZEB1* transcriptional activity was repressed by dephosphorylation of Rb, a tumor suppressor protein, which is phosphorylated and inactivated by cyclin dependent kinases (CDKs) in most cancer types [134], which then block EMT of highly invasive HT1080 cells in 3D culture [135].

Communication between the malignant epithelial cells and the surrounding stromal cells such as fibroblasts or inflammatory cells is known to play a key role in the metastasis and invasion of BC [136]. Although several studies report a change in the gene expression profile of BC cells co-cultured with fibroblasts [137,138], experiments performed in 2D monolayer culture fail to recapitulate cell–cell interactions and the natural 3D cell environment. The progression and advancement in the application of 3D co-culture systems provide more physiologically relevant information regarding EMT and metastasis than 2D cultures. For example, osteocytes, the most abundant cells in the bone matrix, and the osteocyte-secreted bone matrix altered adhesive and migratory behaviors of tumor cells in a 3D co-culture model and caused compaction and shrinkage of BC cells, which eventually terminated the EMT and allowed the attachment of metastatic BCs to the bone microenvironment through dysregulation of Snail [139]. Another study questioned whether EMT leads to a more migratory phenotype or brings about a more targeted migration profile. They demonstrated that TGF-β-induced EMT is dependent on CCR7 for targeted dissemination towards the lymphatic system in a mouse model and in a 3D co-culture system [140].

## 4. Transcriptional Regulation of Apoptosis

As discussed above, modelling mammary acini morphogenesis and BC pathogenesis in 3D cell culture systems provides an opportunity to study the molecular mechanisms responsible for normal epithelial glandular architecture and how this polarized architecture is disrupted in BC cells. Apoptosis, known as a programmed cell death mechanism, is essential for cellular and organismal homeostasis and survival and has previously been implicated in the formation and maintenance of a hollow lumen and polarized structure during normal mammary morphogenesis [72]. This polarized and well-organized structure of epithelial cells has a critical function for maintaining a normal, differentiated phenotype and for controlling cell proliferation and survival [141]. Studies have shown that BIM, a pro-apoptotic BH3-only BCL family protein, is necessary for increased apoptosis in order to maintain the hollow architecture of the MCF10A cell line during morphogenesis [142]. As described above, disruption of the intact and well-ordered glandular architecture and lack of a hollow lumen are hallmarks of glandular epithelial tumors. Therefore, tumor cells must evade luminal apoptosis to elicit lumen filling. Oncogenes, such as mutant forms of SRC or RAS, and activated ErbB2 (HER2/neu) induce constitutive proliferation and luminal filling via suppression of BIM expression [143].

Acquired resistance to apoptosis is often considered a hallmark of cancer [136]. During cancer development and progression, stress conditions, such as increased oncogenic activity, DNA damage, or detachment from ECM, trigger apoptosis. Detachment from the ECM is a key step especially in metastasis, but since epithelial cells require attachment to the ECM, this induces cell detachment-induced apoptosis called anoikis (Greek word for “homelessness”) [144]. Anoikis, first reported in 1994, can be defined as an alternative programmed cell death mechanism that is induced via disruption of the interactions between normal epithelial cells and ECM [145]. Thus, cancer progression requires resistance to anoikis to provide viability of ECM-detached metastatic cells. (Anoikis is out of the scope of our article, and there are extensive review articles about it [146,147,148]. Several studies performed in 2D culture models have demonstrated that metastatic cells gain anoikis resistance through alterations of their receptors and effectors [149,150]. In this regard, receptor tyrosine kinases (RTKs) such as IGF1R, EGFR, PTK6, and ErbB2 are usually dysregulated in cancer with respect to promoting proliferation through anchorage-independent growth. In addition, small GTPases inhibit anoikis by regulating different types of intracellular processes. Mutations in HRAS, KRAS, and NRAS are related to aberrant cancer cell survival and can also enhance the anchorage-independent growth. Aberrant expression of TFs that are related with aggressiveness such as Twist, Slug, and SMAD3 can also confer resistance to anoikis [151,152]. Here, we will provide an overview of TFs regulating apoptosis and anoikis mechanisms in 3D culture models of BC.

Transcription factor NFKB, a known modulator of cell survival, drives apoptosis resistance in mammary epithelial cell acini. Resistance to apoptosis is acquired through laminin-induced ligation of β4 integrins followed by hemidesmosome formation and endogenous activation of NFKB in non-polar mammary epithelial cells [56].

Emerging 3D cell culture methods accelerate the research on anoikis and anoikis-related pathways. It has been shown that ErbB2 activation stimulates cell proliferation, disrupts the polarized organization of epithelial cells, and forms structures containing multiple acinar units in growth-arrested polarized mammary acini cultured in 3D BM gels [143]. Moreover, ErbB2 downregulates Irf6, a member of the interferon-regulatory factor family of transcription factors, in HER2-positive BC cells to block anoikis and promote anchorage-independent growth by activating MAP kinases that inhibit transcription factor ∆Np63α-dependent signals required for Irf6 upregulation [153].

## 5. Transcriptional Regulation of Cancer Therapy Resistance and Drug Response

Due to the heterogeneous nature of BC, chemo- and radio-resistance and differential drug responses are considered the major challenges influencing the overall survival and relapse rates in treatment. Over time, extensive literature has been accumulated about the effect of tumor heterogeneity and the TME on molecular mechanisms of drug response and chemo- and radio-resistance [154,155]. The TME challenges drug response by generating an environmental barrier that prevents drug accumulation [156]. Additionally, dynamic signalling interactions between tumor cells and stroma in the TME promote de novo drug resistance called “environment-mediated drug resistance”, which transiently protects cells from anti-cancer therapy-mediated apoptosis [157,158]. Studies using 2D cultured cancer cell lines are unable to recapitulate most of the in vivo features of tumors such as ECM-to-cell signalling, limited oxygen supply, nutrient gradients, cell–to-cell interactions, variable rates of cell proliferation, and impact of stromal cells in the TME. Therefore, it is now being recognized that drug activity tends to be overestimated in 2D-culture-based platforms [159]. This is further supported by the fact that numerous anticancer agents often fail in clinical studies [160]. In contrast to 2D monolayer culture models, 3D cell culture models are shown to be better at representing the in vivo TME, which is especially evident with the observed inconsistency and variability in drug response and resistance of different cancer cell types when evaluated in a 3D culture system compared to 2D [161]. For example, MCF7 cells grown in 3D culture were found to be more resistant to doxorubicin, paclitaxel, and tamoxifen than those grown in traditional 2D culture [162,163]. In contrast, an increased response to trastuzumab (Herceptin), a monoclonal antibody specifically targeting HER2, was detected in the SKBR3 cell line upon culturing in a 3D culture [164]. These results underline the significant contribution of tissue architecture and ECM in drug response and chemoresistance. Thus, the information acquired from 3D cell cultures provides a more accurate representation of drug activity and response seen in vivo. In this section, we will summarize the literature focusing on TFs regulating chemoresistance and drug response in 3D culture models of BC.

The 3D cell culture models can mimic the hypoxic condition caused by the TME, which is among the hallmarks of solid tumors in vivo. Consistent with the tumor cells in solid tissues, the hypoxic status of 3D structures, i.e., spheroids, is determined by their size, which affects the distribution of oxygen and nutrients [165]. Indeed, oxygen concentration and cell viability decrease and the hypoxic zone increases as a function of spheroid size. Earlier studies have shown that small spheroids with a size of <200 μm diameter mainly consist of proliferating and normoxic cells and do not develop a hypoxic zone in the core [166,167,168]. Different approaches may thus be considered to resemble pathophysiological conditions with hypoxic zones. For example, the scaffold-based 3D culture of MCF10A cells developed into small acinar structures of <200 μm [169], but a scaffold-free hanging drop method enabled the generation of highly spherical, large (~1 mm diameter), and self-organized 3D structures of MCF10A cells [170].

In a tissue environment that lacks oxygen supply, tumor cells adapt their metabolism via induction of hypoxia-inducible factor (HIF)-1α, a central TF mediating hypoxic adaptation and tumor growth through various mechanisms including glucose metabolism, angiogenesis, cell survival, and apoptosis [171]. There is also a consensus that hypoxia promotes chemoresistance in several tumor types including BC, mainly through HIF1α-mediated transcriptional regulation of many genes that control drug resistance [172]. For example, HIF-1α was shown to be activated in MCF7 cells cultured in a 3D culture resembling the in vivo hypoxic nature of the TME [173]. These cells were more resistant to doxorubicin compared to MCF7 cell monolayers. Indeed, HIF-1α interacts with the promoter region of the *MDR1* (multidrug resistance 1) gene, thereby controlling its transcriptional regulation to mediate doxorubicin resistance [173].

Constitutive activation of NFKB in tumors including BC has been directly linked with resistance to chemotherapy and radiation [174,175]. In a research article published by Ahmet et al., this phenomenon has been studied in a 3D culture of BC cells, suggesting a loop-like activation pathway of NFKB/β1-integrin signalling. In fact, NFKB activation was found to be required for radiation-induced β1-integrin overexpression through increased transcriptional activity, leading to tumor radioresistance in malignant breast cells [176]. Previous studies showed that cancer cells have a constitutive activation of NFkB due to overexpression of tissue transglutaminase, TG2 [177]. In addition to its stimulatory effect on cell mobility, invasion, and metastasis, which is mainly driven through EMT induction, TG2 can also induce chemoresistance by acting on the structure and stability of the ECM by crosslinking fibronectin and enabling ECM to protect cancer cells from exposure to chemotherapy [178]. Kumar et al. has provided evidence for the link between TG2 and metastasis and drug resistance [179]. A 3D culture of mammary epithelial cells showed that TG2-induced activation of NFKB is responsible for transcriptional regulation of Snail1 and loss of both cell–to-cell interaction and BM integrity. Furthermore, mammary epithelial cells undergoing TG2-inducing EMT became resistant to doxorubicin-induced cell death [179].

Prior research by Bihani et al. has emphasized that MYC is a driver of everolimus resistance in ERα(+) BC cells and in long-term estrogen-deprived models that mimic patient tumors with anti-estrogen resistance [180]. Upregulation of *MYC* expression upon either everolimus resistance or acute everolimus treatment was shown to be mediated by its increased association with BRD4 (Bromodomain-containing protein 4), which is clinically associated with multiple types of cancer by transcriptionally and epigenetically regulating key oncogenic drivers [181]. The combinational therapy involving both everolimus and the BRD4 inhibitor resulted in increased growth inhibition compared with everolimus treatment alone in 3D culture and in an in vivo xenograft model of MCF7 [180].

Novel targets that potentiate statin-induced apoptosis were identified by genome-wide RNAi screening in BC cells [182,183]. Genes in the mevalonate pathway including *SREBP2* that are involved in tumor development, progression, and drug response [184,185,186] were among the top-scoring results. Stable knockdown of *SREBP2* alone did not impair the rate of cell growth in 3D, yet MDA-MB-231 and MCF7 cells were sensitized to the pro-apoptotic and anti-proliferative effects of fluvastatin through *SREBP2* knockdown [187].

A novel method called TRACER (TRanscriptional Activity CEll aRrays) was reported in the literature enabling quantification of the dynamic activity of numerous TFs simultaneously in 3D culture [188]. The method has been applied to ErbB2-positive BC cells and patient samples to identify TFs that are responsible for the mechanism of action of lapatinib, a targeted therapeutic agent against ErbB2. Hence, the activity of a number of TFs in 3D culture including E2F1, ELK1, GATA, P53, and STAT4 was found to be increased upon lapatinib treatment, which suggests their contribution to the mechanism of action of lapatinib [188].

## 6. Transcriptional Regulation of Stemness

Advances in sequencing technologies have subsequently provided a great depth of information regarding tumor heterogeneity. Among different cell populations of the same breast tumor tissue, breast cancer stem cells (BCSCs) are described as a dynamic subpopulation of cancer cells that have the capability of self-renewal and differentiation [189]. Hence, BCSCs account for several key processes during tumorigenesis including self-renewal, tumor initiation and maintenance, metastasis, therapy resistance, and relapse [190]. Previous studies have emphasized that cancer stem cells (CSCs) are dependent on and are regulated by complex interactions with the TME [191]. Considering the role of CSCs in tumorigenesis, identification of the key factors and related signalling pathways involved in BCSCs maintenance is highly critical. However, difficulties in culturing CSCs in traditional 2D monolayer culture is the main constraint in the study of CSC biology. In contrast, CSCs studies in 3D cell culture models contribute further to our understanding of the role of CSCs in tumorigenesis since these models can successfully mimic in vivo TME that is required for CSC enrichment and proliferation [192]. This section reviews the literature related to the TFs involved in BC stemness in 3D cell culture models.

A recent study by Yu et al. suggested that BC stemness can be regulated through the coordination of two TFs, SOX2 (Sry-related high-mobility box 2) and FOXO1 (Forkhead box O1), which together elevate the expression of *TRIB3* [193]. TRIB3 acts as a stress sensor and has been shown to correlate with poor prognosis in BC patients [194]. According to their proposed mechanism, TRIB3 supports BC stemness by stabilizing FOXO1 by halting its degradation and leading to FOXO1-dependent transcriptional expression of *SOX2* to maintain BC stemness. Consistently, TRIB3 promoted the formation of the tumorsphere in 3D culture, and silencing of *SOX2* significantly impaired TRIB3-promoted tumorsphere formation. Overall, the TRIB3-FOXO1-SOX2 axis was introduced as a novel mechanism for supporting BC stemness in 3D cell culture models [193]. In another study, repression of oncogene *Tbx3* [195,196] by the tumor suppressor miR-206 [197] in MDA-MB-231 cells caused a significant loss of growth in 3D culture by inhibiting the colony size, branching morphology, and invasiveness of a BC stem cell population [198].

Additionally, stemness properties of BC cells were studied in 3D mammosphere models with two luminal BC cell lines, MCF7 and SKBR3, and a basal BC cell line, MDA-MB-231, to identify potential common regulatory TFs. At both mRNA and protein levels, expressions of Nanog and Oct4 markers were increased in 3D mammospheres in comparison with the 2D monolayer counterparts [199]. It was hypothesized that Glycoprotein nmb (GPNMB), which is a type I transmembrane protein [200], could be related to the acquisition of CSC-like properties in BC [201]. To evaluate initiation of BCSC properties, 3D spheres and 2D monolayer models have been established with three different BC cell lines as BT-474 (luminal), Hs578T (basal), and MDA-MB-468 (basal). The expression level of Glycoprotein nmb on the cancer cell surface was found to be correlated with CSC properties. More importantly, the difference between the CSC phenotypes and intracellular gene regulation of stemness between 2D and 3D culture models were examined. When cultured in 3D, stemness markers such as SOX2, NANOG, OCT4, and FOXO3 had higher expression levels compared with 2D culture. Therefore, this study provides another scientific example that the 3D models have a greater potential for investigating the BCSC content and stemness character of the tumor samples than traditional 2D models [201].

## 7. Conclusions

Being the standard for in vitro cell–based studies for decades, traditional 2D cell culture models significantly improved our knowledge of cancer biology and have provided a wealth of reference literature. Despite the advantages of well-established and proven 2D culture models, they lack 3D spatial cellular distribution and organization of solid tumors, which limits these models to reflect in vivo properties of cancer. By contrast, 3D in vitro systems, which have become the next frontier of cancer research, can overcome some of these limitations. [36,202,203]. In this context, biomaterials of both natural and synthetic origin are increasingly being exploited to develop 3D models [204]. Natural polymers (e.g., Matrigel^®^, collagen and collagen-derivatives, fibrin, hyaluronic acid, alginate, and chitosan) are highly cytocompatible and better-mimicking native 3D tissue but with moderate control of system tunability. In contrast, synthetic polymers (e.g., PEG, PLGA) have adjustable biophysical parameters such as hydrogel stiffness and matrix permeability [205]. A wide range of 3D platforms has been manufactured for BC cell culture including scaffold-free spheroids, scaffold-based systems, microfluidic devices, and 3D bioprinting [206]. For example, an early stage of breast tumor invasion has been modeled via embedding multicellular tumor spheroids of MDA-MB-231 and human mammary fibroblasts (HMF) into a 3D hybrid hydrogel system composed of collagen and alginate with adjustable stiffness and high matrix permeability [207]. This biomimetic breast TME provided insight into chemokine signaling implicated in BC metastases. Another breast tumor invasion platform is a gelatin hydrogel microsphere (GM)-based 3D coculture system of CAF or tumor-associated macrophages (TAM) with MCF7 cells enabling effective evaluation of the in vitro invasion of BC cells [208]. How the tumor stroma controls the growth and aggressiveness of BC cells was addressed through a Matrigel-based co-culture system [209]. Malignant fibroblasts increased proliferation of co-cultured luminal but not basal-like BC cells in the absence of estrogens through PDGF ligands secreted by the malignant stromal cells. Furthermore, an advanced microfluidic 3D in vitro model has been proposed as a suitable platform for modeling metastases of MDA-MB-231 cells into a bone-mimicking microenvironment [210]. The development of in vitro miniaturized model systems of organs (named “organoids”) by Hans Clevers’ and many other labs represents a huge breakthrough in translational cancer research involving tissue physiology and disease modeling [211]. A highly efficient and standardized protocol to generate 3D organoid models from primary breast tumor tissues was developed by Sachs et al. [212]. These models preserve the original tumor status including histopathology, hormone receptor status, DNA copy number variation, and the mutational landscape representing the molecular and phenotypic diversity of BC and therefore present an enormous potential in drug discovery and personalized medicine. Furthermore, de novo oncogenesis of BC was modeled through CRISPR-Cas9-mediated targeted knockout of frequently mutated tumor suppressor genes (*P53*, *PTEN*, *RB1*, *NF1*) in organoids derived from human breast epithelial cells [213]. Despite the high degree of the physiological relevance of organoid technology, similar to any other system, it has its own limitations: Along with being a highly variable and heterogeneous system due to clonal or inter-patient variability, there remains a lack of standardization in 3D culture maintenance and further downstream analysis [214]. Moreover, limited optical accessibility of these models often impairs organoid mechanobiology studies [215]. Most critically, they are prone to diverge from their original tissue status and lose their cellular identity over time [215].

In this review, we gave insight into 3D cell culture models that represent the complexity of BC at the molecular level and focused on the TF repertoire of distinct biological events. Section 1 covers the effect of TME on the breast cell morphology and on the process of disorganization observed during tumorigenesis, which can be modeled by taking advantage of 3D models to pinpoint the potential regulatory networks. In addition, 3D models mimic physiologically relevant TME required for EMT, metastasis, and invasiveness, which is not possible to study in traditional 2D models. Studies mentioned in Section 2 clearly demonstrate that ECM elements preserve the integrity of tumor–stroma interplay, and co-culture systems of two or more types of cells including tumor cells, immune cells, and endothelial cells have a vital role in our understanding of the mechanisms controlling each step of metastasis from initiation to termination. Furthermore, resistance to apoptosis and anoikis is regulated by signals from the TME, and gene regulatory networks are responsive to these signals that reprogram the cell fate (Section 3). Section 4 centers around the effect of TME on the drug response; hence, the complexity and heterogeneity of TME also affect both the drug accumulation at the site of the tumor and drug penetration into the disorganized tumor mass. Monolayered 2D models are inefficient to reflect not only the effect of TME but also the penetration process into tumor mass, whereas 3D models efficiently predict the drug response and identify cancer therapy resistance mechanisms influenced by tissue architecture and the TME. In the last section of this review, it is shown that 3D models recapitulate stem-cell niches, which enable maintenance of CSCs through a dynamic interaction with the surrounding cells, which leads to better evaluation of the stemness character of BC (Section 5). Reports clearly showed that there is a differential regulatory program between 2D and 3D models to evaluate the stemness character of BC. TFs that regulate the genes involved in the above-mentioned processes in 3D culture models are listed in Figure 3 and Table 2.

Considering the studies specified in each section of this review article, there is no doubt that the applicability of 3D cell culture models will open new avenues for cancer research, and the transition from traditional 2D culture models to 3D culture models is inevitable in the near future to better reflect the in vivo properties of cancer.

## Figures and Tables

**Figure 1 cancers-14-01023-f001:**
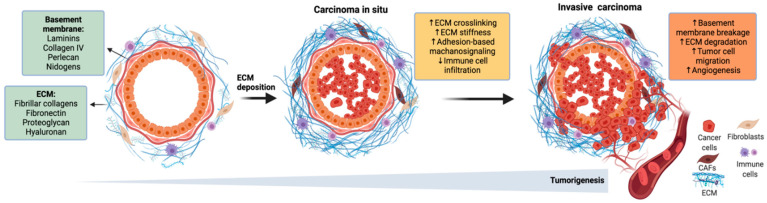
ECM remodeling in breast cancer progression and metastasis.

**Figure 2 cancers-14-01023-f002:**
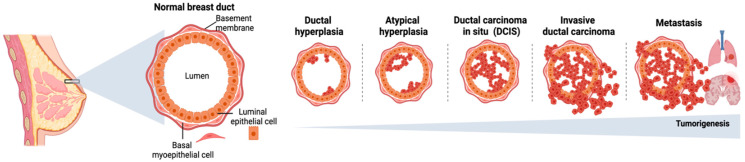
Subsequent steps of tumorigenesis in breast cancer, from ductal hyperplasia to the formation of metastasis.

**Figure 3 cancers-14-01023-f003:**
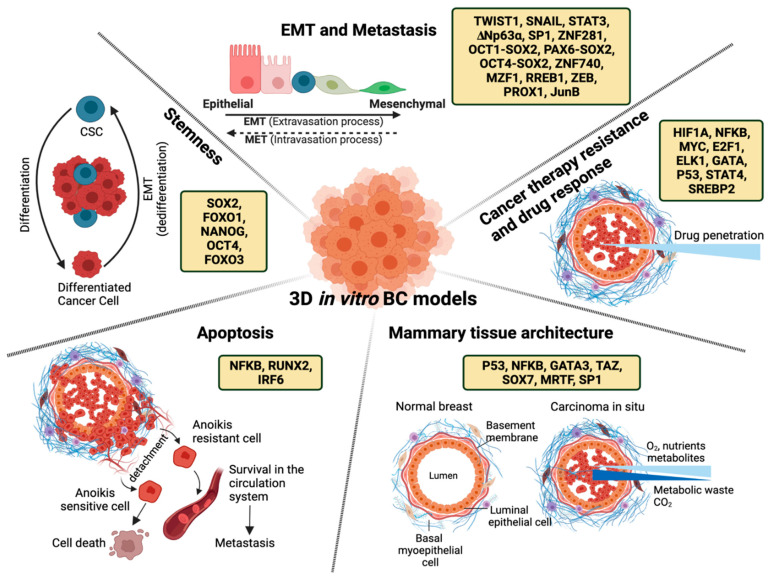
Transcription factors regulating mammary tissue architecture, EMT and metastasis, apoptosis, stemness, and drug response in 3D in vitro breast cancer models.

**Table 1 cancers-14-01023-t001:** Differences between 2D and 3D cell culture systems.

Characteristics	2D Culture	3D Culture	References
Cellular morphology	Cells cultured as monolayers; changed morphology; loss of epithelial cell polarity and diverse phenotype	Cells cultured in 3D aggregates, with preserved original morphology and polarization of cells; diverse phenotype	[33]
Proliferation	Often proliferate more rapidly than in vivo	Depends on 3D matrix interactions	[34]
Differentiation	Poorly or moderately differentiated	Well differentiated	[35]
Cell interactions	Limited cell–cell and cell–microenvironment interactions	Cell junctions enable cell communication	[36,37]
Gene expression	Cell adhesion-, proliferation-, and survival-related genes are usually modified	Similar gene expression profiles to those in vivo	[38]
Drug sensitivity	More sensitive to drugs, high efficacy Drugs are not well metabolized	More resistant to drugs, low potency Enhanced drug metabolism	[39]
Response to stimuli	Limited representation of the response to mechanical and biochemical cues	Accurate representation of in vivo stimuli	[40,41]
Cell exposure to medium	Equal access to nutrients and growth factors	Differential nutrient and growth factor availability, mimicking hypoxic conditions, very similar to in vivo tissues	[42]
Cost	Inexpensive, suitable for large-scale studies, commercially available materials	More expensive than 2D culture, time consuming	[43]
Quality	High performance, reproducible, easily interpretable, suitable for long-term culture	Worse performance and reproducibility, difficult to interpret, difficult to maintain long-term culture	[43]

**Table 2 cancers-14-01023-t002:** Transcription factors (TFs) in BC pathogenesis studied in 3D culture.

Transcriptional Regulation of	TFs	Target Gene	Proposed Function in breast Carcinogenesis	Cell Lines Used for 3D Culture	Type of ECM	References
Mammary tissue architecture	P53	SREBP-1 and SREBP-2	Disruption of normal acinar morphology via mutant p53	MDA-468, MDA-231, and MCF10A	rBM matrix	[79]
	NF-κB	Disorganization genes	Disruption of tissue polarity and organization	HMT-3522 S1 and T4-2 cells	rBM matrix	[83]
	GATA3		Induces proliferation via mutant GATA3s	ZR751	Xenograft	[87]
	TAZ	IRS1	Induces proliferation	MCF10A	rBM matrix	[90]
	SOX7		SOX7 depletion causes loss of polarity and increased proliferation	MCF10A	rBM matrix	[93]
	TBX3		Branching morphology	MDA-MB-231	rBM matrix	[198]
	MRTF-A and MRTF-B	integrin α5, integrin α6	Disruption of normal acinar morphology	MCF10A	rBM matrix	[96]
	Sp1		Stiffness-induced tumorigenic phenotype and chromatin organization	MCF7 and MDA-MB-231	rBM matrix	[100]
EMT and metastasis	TWIST1	ADAM12	Induces tumor invasion and metastasis	Hs578T and HMLE cells	Matrigel–collagen mixture	[113]
	TWIST1	PRKD1	Promotes invasion, epithelial dissemination, and metastasis	murine and human mammary tumor tissues	Organoid	[116]
	SNAIL		SNAIL inhibition reduces invasion and metastasis	MCF7		[120]
	SNAIL		Downregulation of Snail induces a reversal of EMT in the bone	MDA-MB-231, mammary tumor cells, osteoblast-like cells	rBM matrix	[139]
	STAT3	LINC00520	Invasion and metastasis	MCF10A-Src transformed cells	rBM matrix, bioprinting	[121]
	STAT3		STAT3 inhibition blocks invasion	primary tumor tissue explant culture	rBM matrix	[122]
	∆Np63α		∆Np63α inhibition induces EMT	MCF10A	3D collagen gel	[126]
	OCT1-SOX2, PAX6-SOX2, OCT4-SOX2		aggressive cancer phenotype	T47D		[130]
	SP1, ZNF281, ZNF740, MZF1 and RREB1		aggressive cancer phenotype	T47D		[130]
	ZEB1	E- and N-cadherins	Induces EMT	HT1080	rBM matrix	[135]
	PROX1	MMP14	Inhibits invasion	MDA-MB-231	3D fibrin matrix	[127]
	JunB		EMT-induction	NMuMG and iLEC cells	3D co-culture fibrin beads	[140]
Apoptosis	NF-κB		Induced by tissue polarity and drives resistance to apoptosis	HMT-3522 sourced S-1 and T4-2 cells	rBM matrix	[56]
	RUNX2	Bcl-2 and IL-8	Disruption of normal acinar morphology, increased cell proliferation	MCF10A and MDA-MB-231	rBM matrix	[91]
	IRF6		Induced by detachment and downregulation causes anoikis resistance	MCF10A, HMEC	rBM matrix	[153]
Therapy resistance and drug response	HIF-1α	MDR-1	Increased resistance to doxorubicin	MCF7		[173]
	NF-κB	β1-integrin	Increased radioresistance	HMT-3522 S1 and T4-2 cells	rBM matrix	[176]
	MYC		Drives resistance to everolimus	MCF7, ZR75, CAMA-1	rBM matrix	[180]
	E2F1, ELK1, GATA, P53 and STAT4		Increased activity upon lapatinib treatment	MCF10A/ErbBB2	rBM matrix	[188]
	SREBP2	HMGCR and HMGCS1	Sensitizes to fluvastatin-induced apoptosis	MDA-MB-231 and MCF7	rBM matrix	[187]
Stemness	SOX2 and FOXO1		Accumulation of FOXO1 and enhanced SOX2 transcription supports BC stemness	Human breast cancer tissues	ULA plates	[193]
	NANOG and OCT4		Increased expression in mammospheres	MCF7 and SKBR3, and MDA-MB-231	Agar-coated plates	[199]
	SOX2, NANOG, OCT4, and FOXO3		Increased expression in cell surface-GPNMBhigh cells	BT-474, Hs578T, and MDA-MB-468	ULA or HEMA-coated plates	[201]

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
