# Peer review of "Transcriptional Factor Repertoire of Breast Cancer in 3D Cell Culture Models"

_cancers, 2022, doi:10.3390/cancers14041023_

Round 1
Reviewer 1 Report
This is a review for the manuscript by Ozkan et al titled “Transcriptional factor repertoire of breast cancer in 3D cell culture models.” This is a comprehensive review article that is well written, well researched, and overall, a high quality manuscript. This reviewer has only these very minor concerns:
Classifications of breast cancer have expanded beyond luminal A, luminal B, HER2+ and basal-like. A line acknowledging this should be added to the first paragraph of the Introduction.
Line 216-fibroblasts are listed twice
A few spelling and grammar mistakes were observed.
Reviewer 2 Report
This paper is a review of 3D breast cancer models with transcriptional factor repertoire. This review is well-written and valuable for tissue engineering or cancer biology researchers. However, some sentences or sections are lacking. For example, the authors should prepare the table to describe the characteristics of transcriptional factor repertoire. In addition, to construct and understand the 3D models, biomaterials are crucial. The description is needed for readers’ better understanding. Taken together, major revisions should be made before re-submission. The paper would be accepted only when the below comments are responded.
The authors should add the sentences for the role of the biomaterial-based 3D cancer model. The field is essential in this review for readers’ better understanding. It has been reported that biomaterials support EMT induction. To reduce the authors’ burden, I suggest at least these recent papers be added for revision (review and research paper).
Review papers
Cancers 2020, 12(10), 2754.
Tissue Engineering Part B: 2010. 351-359.
Research papers (Especially gene or EMT-related research)
Acta Biomater. 2018, 75, 213–225.
Tissue Eng. Part A, 26, 2020, 1272-1282. https://doi.org/10.1089/ten.tea.2020.0095
BMC Cancer 2014, 14, 735.
Biomaterials 35 (2014) 2454-2461
The authors should add the table for the TFs.
Written English of the manuscript should be considerably improved, and therefore, a revision by a native speaker is highly recommended.
Example (Abstract)
Moreover, ECM and tissue architecture profoundly impact chromatin structure, and thereby on gene expression.
Author Response
This paper is a review of 3D breast cancer models with transcriptional factor repertoire. This review is well-written and valuable for tissue engineering or cancer biology researchers. However, some sentences or sections are lacking. For example, the authors should prepare the table to describe the characteristics of transcriptional factor repertoire. In addition, to construct and understand the 3D models, biomaterials are crucial. The description is needed for readers’ better understanding. Taken together, major revisions should be made before re-submission. The paper would be accepted only when the below comments are responded.
We would like to thank Reviewer 2 for the useful comments and suggestions that improved our manuscript significantly. Especially, the addition of an extra paragraph discussing the role of the biomaterial-based 3D cancer model is definitely improved the depth of our manuscript.
Please find below our responses to the Reviewers’ comments, which details the changes we made to the manuscript and the concerns that we have addressed as follows:
Point 1: The authors should add the sentences for the role of the biomaterial-based 3D cancer model. The field is essential in this review for readers’ better understanding. It has been reported that biomaterials support EMT induction. To reduce the authors’ burden, I suggest at least these recent papers be added for revision (review and research paper).
Review papers
Cancers 2020, 12(10), 2754.
Tissue Engineering Part B: 2010. 351-359.
Research papers (Especially gene or EMT-related research)
Acta Biomater. 2018, 75, 213–225.
Tissue Eng. Part A, 26, 2020, 1272-1282. https://doi.org/10.1089/ten.tea.2020.0095
BMC Cancer 2014, 14, 735.
Biomaterials 35 (2014) 2454-2461
Response 1: We agree with the Reviewer on the importance of the role of biomaterial-based 3D models in cancer research. Following the Reviewer’s suggestion, we have now incorporated a comprehensive paragraph about this topic in the Conclusion part and discussed each review and research paper suggested by the Reviewer as follows: “In this context, biomaterials of both natural and synthetic origin are increasingly being exploited to develop 3D models [204]. Natural polymers (e.g., Matrigel®, collagen and collagen-derivatives, fibrin, hyaluronic acid, alginate, and chitosan) are highly cytocompatible and better mimicking native 3D tissue but with moderate control of system tunability. In contrast, synthetic polymers (e.g., PEG, PLGA), are with adjustable biophysical parameters such as hydrogel stiffness and matrix permeability [205]. A wide range of 3D platforms has been manufactured for BC cell culture including scaffold-free spheroids, scaffold-based systems, microfluidic devices, and 3D bioprinting [206] . For example, an early stage of breast tumor invasion has been modeled via embedding multicellular tumor spheroids of MDA-MB-231 and human mammary fibroblasts (HMF) into a 3D hybrid hydrogel system composed of collagen and alginate with adjustable stiffness and high matrix permeability [207]. This biomimetic breast TME provided an insight into chemokine signaling implicated in BC metastases. Another breast tumor invasion platform is a gelatin hydrogel microsphere (GM)-based 3D coculture system of CAF or tumor-associated macrophages (TAM) with MCF7 cells enabling effective evaluation of the in vitro invasion of BC cells [208]. How tumor stroma controls growth and aggressiveness of BC cells was addressed through a Matrigel-based co-culture system [209]. Malignant fibroblasts increased proliferation of co-cultured luminal but not basal-like BC cells in the absence of estrogens through PDGF ligands secreted by the malignant stromal cells. Furthermore, an advanced microfluidic 3D in vitro model has been proposed as a suitable platform for modeling metastases of MDA-MB-231 cells into a bone-mimicking microenvironment [210].” (pg 14 and lines 726-747, references 205-210)
Point 2: The authors should add the table for the TFs.
Response 2: In fact, we have already included a table summarizing all the transcription factors discussed in our manuscript. Please check Table 2, pg 17-20.
Point 3: Written English of the manuscript should be considerably improved, and therefore, a revision by a native speaker is highly recommended.
Example (Abstract)Moreover, ECM and tissue architecture profoundly impact chromatin structure, and thereby on gene expression.
Response 3: We have now fixed this inaccuracy in the revised manuscript. As it was suggested by the Reviewer, the text was revised and reviewed by a native English
Reviewer 3 Report
In this manuscript, Özkan et al. provide an interesting overview of transcription factors activity in 3D models of breast cancer. After an initial comparison between 2D and 3D cell culture methods, they specifically focus on transcription factors that have been shown to act differently in cell monolayers and in 3D environments. Research on transcription factors is divided by areas of interest, starting with breast tissue architecture, then EMT, apoptosis, drug response, and stemness.
I find the manuscript clear and I especially like the choice of dividing the body of the text into the previously mentioned sections.
I have only a few comments:
- The authors extensively cite the woks of the Bissell lab, sometimes forgetting others whose contribution to the field has been not negligible. I agree that Mina Bissell has been a pioneer of 3D models, but today many laboratories all over the world have contributed to this expanding field. The Clevers lab and the Weaver lab (the former being an alumna of Bissell), just to mention some, deserve to be cited as well.
- I agree with the authors that 3D models better recapitulate the in vivo tumors, but I would tone down the introduction because as of now, it seems that all the knowledge acquired from 2D cell culture has to be discarded (and the authors cite plenty of work done in 2D). I would try to make a more objective comparison between the two systems, citing also some advantages of 2D cultures (for example: possibility to perform super resolution microscopy). With regards to this point, I would like to draw the attention of the authors to a recent paper (Pérez-Gonzalez et al. 2020 https://doi.org/10.1038/s41556-021-00699-6) where intestinal organoids were “melted” back to a monolayer and cells retained their compartmentalization.
- Following the previous point, the authors say that EGFR-integrin crosstalk is not observed in 2D. This is not true as shown by several repors (for example: https://doi.org/10.1038/onc.2011.107 and https://doi.org/10.1038/ncb1209)
- In Table 1, the authors state that cell-cell interactions are lacking in 2D. I think this is probably an overstatement.
- Still in Table 1, the authors mention that cells in 2D have a poor response to mechanical and biochemical cues. This is not clear to me, especially for mechanical responses. In fact, many mechanobiology researches have been performed only in 2D.
- When the authors talk about TP53 (lines 248-262), it could be of interest to briefly discuss this work from Norman’s lab about p53 modifying the ECM via exosomes: 1038/s41467-018-07339-y
- In the part relative to hypoxia, the authors state that 3D cell culture models precisely mimic the hypoxic conditions (line 523). This is definitely true for big, millimeter-sized organoids. I am not sure that small organoids such as those one can obtain with CACO2, MDCK, or MCF10A cells in matrigel are sufficiently big to generate an oxygen gradient. Can the authors comment on this?
- Minor comment: line 468, the authors probably meant to say “out of the scope”.
Round 2
Reviewer 2 Report
The authors have responded to all the comments. I recommend the publication.